# Design Optimization of Sensitivity-Enhanced Structure for Fiber Bragg Grating Acoustic Emission Sensor Based on Additive Manufacturing

**DOI:** 10.3390/s22020416

**Published:** 2022-01-06

**Authors:** Yang Yu, Bo Liu, Feng Xia

**Affiliations:** School of Information Science and Engineering, Shenyang University of Technology, Shenyang 110870, China; yuy@sut.edu.cn (Y.Y.); xiafeng@sut.edu.cn (F.X.)

**Keywords:** fiber Bragg grating, additive manufacturing, acoustic emission, FEA, improving sensitivity

## Abstract

A four-loop shaped structure of fiber Bragg grating (FBG) acoustic emission (AE) sensor based on additive manufacturing (AM) technology is proposed in the letter. The finite element analysis (FEA) method was used to model and analyze the sensor structure. We aimed at improving the sensitivity, the static load analysis, and the dynamic response analysis of the normal FBG acoustic emission sensor and the FBG AE sensor with improved structure parameters. We constructed the FBG AE sensor experimental system based on a narrowband laser demodulation method and test on real acoustic emission signals. The results demonstrated that the response sensitivity of the FBG acoustic emission sensor was 1.47 times higher than the sensitivity of the normal FBG sensor. The sensitivity coefficient of PLA-AE-FBG2 sensor was 3.057, and that of PLA-AE-FBG1 was 2.0702. Through structural design and parameter optimization, the sensitivity and stability of the FBG AE sensor are improved. The four-loop shaped sensor is more suitable for the health monitoring in fields such as aero-engine blade, micro-crack of structure, and crack growth in bonded joints. While ensuring the sensing characteristics, sensitivity, and stability of the four-loop shaped sensor have been enhanced. It is possible to apply the FBG AE sensor in some complex engineering environments.

## 1. Introduction

Acoustic emission (AE) refers to the phenomenon of transient elastic waves that are emitted by materials due to the rapid release of energy. As an important nondestructive detection technology, AE detection has been successfully used in the field of structural health monitoring and early material flaw detection [1]. In some applications of strong electromagnetic interference and narrow application environment, such as high-voltage transformers and aero-engine blade damage monitoring, the application of traditional AE sensors is greatly limited due to their large size and susceptibility to electromagnetic interference and other factors.

A fiber grating sensor has the advantages of rapid response, anti-electromagnetic interference, small size, easy installation, and a high reliability, etc. The sensor network of distributed or quasi-distributed sensors is easier to setup. Therefore, the fiber Bragg grating (FBG) acoustic emission (AE) sensor that combined the above-mentioned advantages and the high sensitivity characteristics of an FBG AE sensor can realize distributed detection applications in strong electromagnetic interference, small space, and large structures. In recent years, the FBG AE sensor has been widely used in the aerospace field, composite material detection, and large-scale structural health monitoring [2,3,4,5,6] as well as other fields. The FBG AE sensor is gradually becoming a sensor with great development potential and application value [7].

The bare grating is suitable for the detection of stress, strain, temperature, and AE signal, but its sensitivity is lower than that of AE sensors and it is very thin and fragile. Therefore, in the practical application of AE detection, it is necessary to design a sensitization structure to improve its sensitivity in the detection of structural damage. The sensitivity enhancement structure of FBG sensor [8] protects the bare grating to avoid damage, and even achieve temperature compensation [9,10] or sensitivity adjustability [11] as required. Thus, the sensitivity enhancement structure of FBG AE sensor is the key to its engineering application and becomes a hot research issue in the application field of AE technology [12,13]. Many researchers have studied the applicability of FBG AE sensors in the structural monitoring of aerospace and large-scale buildings. Giurgiutiu directly embedded fiber gratings into aerospace composite materials for monitoring their structural health [14], however, embedding the sensors complicates the composite manufacturing, and the monitoring strategy must be foreseen before the manufacturing process. Chen Ru et al. used FBG to monitor the stress and strain in AM process in real time [15]; Liu et al. used the phase shifted fiber Bragg grating (PS-FBG) to monitor composite materials for long-term use [16]; Tao et al. directly used the FBG acoustic emission sensor to monitor the health conditions of the polymer-bonded explosive (PBX) cracks in the narrow space of polymer explosives [17]; Manterola et al. used FBG as the AE sensor to treat the cracks that appeared at the structural rigid bonding head for a long time [18]; Leitao et al. developed FBG intensity sensors and the sensor was successfully used in the monitoring of central arterial pressure changes over a long period of time [19].

Arnaldo proposed a three-dimensional (3D) displacement sensor that was based on one fiber Bragg grating (FBG), and the sensor measured the axial strain, bending, and torsion with a single fiber Bragg grating in CYTOP fiber simultaneously with good agreement. The relative errors was below 5.5% [20]. Kumar et al. proposed the simultaneous measurement of the strain and temperature using a pair of etched and un-etched fiber Bragg grating (FBG) that was fabricated in a single mode glass optical fiber. The diameter of the fiber cladding was reduced by etching and the temperature and the strain sensitivity was increased to 11.80 pm/°C and 1.2 pm/µε [21].

At present, some common packaging forms mainly include the paste type [22], the embedded type [23], the substrate type [24], the capillary type [25], and the box type [26]. Compared with the paste type, the embedded packaging solution can protect the fiber grating and avoid direct exposure to the working environment, but there is no way to realize the reuse of the fiber grating. Xiujuan [27] et al. optimized the design of the I-shaped FBG and made the metal copper and steel substrate into the I-shaped package, which improved the sensitivity of the fiber grating and allowed the reuse of the fiber grating sensor. Li [28] used the steel tubes of a capillary to encapsulate the FBG sensor, which was applied to the strain and temperature monitoring of the comprehensive laboratory building of Dalian University of Technology. Torres [29] used finite element analysis to establish a sensor package model and studied the accuracy of the sensor after being pasted with the test piece.

The research of the above-mentioned packaging structure has enhanced the reliability of the grating sensor application. However, some sensors are difficult to encapsulate and some have poor stability and repeatability, especially if the sensitivity is not up to the sensitivity index of AE sensors and cannot be used in the applications for acoustic emission monitoring of structural damage.

Additive manufacturing (AM) offers potential solutions when conventional manufacturing reaches its technological limits. These include a high degree of design freedom, lightweight design, functional integration, and rapid prototyping, Nowadays, AM exhibits the capability and capacity for advanced manufacturing and is another potential process to produce freeform geometries. Various industries have been using AM profitably for many years including high-tech industries, such as aerospace, biomedical, and so on [30,31]. There are many kinds of additive manufacturing materials including commonly used ABS material, polycarbonate material, polyamide material, and polylactic acid (PLA) material [32]. Therefore, PLA was selected as the printing material in this paper. The 3D printing process parameters have a great influence on various aspects of the mechanical properties, and the parameters need to be adjusted when making the parts [33]. After the parts are made, a low-cost heat treatment can improve the impact strength of the printed polymer during the post-treatment process [34]. After the parts have been manufactured, the heat-treatment process might be considered as a low-cost post-processing method for improving the impact resistance of a 3D-printed polymer part considerably [35].

This paper proposes an AM-based FBG AE sensor with an enhanced sensitivity structure, which improves the sensitivity and stability of the sensor, has better thermal stability, higher flexural modulus and mechanical properties, and can effectively protect the fiber Bragg grating. The finite element analysis (FEA) method is carried out on the package structure parameters of the FBG acoustic emission sensor and the static stress and dynamic response were analyzed. A test platform was built for the experimental research. Experimental research shows that both the four-loop shaped FBG acoustic emission sensor and the traditional Piezoelectric Transducer (PZT) AE sensor have good AE signal response characteristics. The four-loop shape structure improves the performance of the FBG AE sensor and effectively ensures the stability of the sensor.

## 2. Theoretical Analysis of FBG Sensing

Fiber grating is a sensor device with periodic modulation of refractive index. When it is not affected by changes in the external physical quantities, the effective refractive index of the core in the axial direction can be expressed by Formula (1)
(1)$$n_{eff}\left(z\right)=\bar{n_{eff0}}-\Delta nsin^{2}\left(\frac{\pi }{\Lambda_{0}}z\right)$$

In the formula, $\bar{n_{eff0}}$ is the average effective refractive index of the fiber grating that is not affected by acoustic emission, $\Lambda_{0}$ is the initial period of the Bragg fiber grating, Δn is the maximum change in refractive index, and *z* is the coordinate point along the axis of the grating.

Using fiber grating coupled mode theory calculation, mode coupling occurs when a beam of light passes through the grating and the peak wavelength of the fiber grating reflection spectrum is the central wavelength, which meets Formula (2)
(2)$$\lambda_{R}=2\bar{n_{eff}}\Lambda$$
where $\lambda_{R}$ is the central wavelength of the grating reflected light, $\bar{n_{eff}}$ is the effective refractive index of the fiber core, and Λ is the grating grid period. Under the action of the lamb wave that is generated by acoustic emission, the strain field model along the axial direction of the fiber grating can be expressed by the cosine function of Formula (3)
(3)$$\varepsilon_{AE}\left(t\right)=A_{AE}\cos\left(\frac{2\pi }{\lambda_{AE}}z-\omega_{AE}t\right)$$

$A_{AE}$ means normalized to AE wave amplitude, $\lambda_{AE}\;$ is the wavelength of the acoustic emission that is propagating in the detection material, $\omega_{AE}$ is the angular frequency of the acoustic emission, *z* is the point along the axis of the fiber, and *t* is the propagation time of the acoustic emission wave. When the fiber grating is affected by the external physical quantities, the grating period and refractive index are affected and the wavelength drifts. The amplitude and strain of the drift can be expressed by Formula (4)
(4)$$\Delta \lambda_{R}=\lambda_{R}\left(1-p\right)\varepsilon_{AE}$$
where $\Delta \lambda_{R}$ is the wavelength shift amplitude caused by strain, and *p* is the elastic-optical coefficient of the optical fiber. $\varepsilon_{AE}$ is the strain amplitude. The elastic-optical coefficient of a typical single-mode silica fiber is *p* = 0.22. For a grating with $\lambda_{R}\;$= 1550 nm, the strain sensitivity that is calculated by Equation (4) is 1.22 pm/με. The fiber grating is subjected to the non-uniform stress field of the acoustic emission wave in the axial direction and after the acoustic emission stress wave is modulated, the wavelength changes non-uniformly with the acoustic emission wave.

## 3. Structural Design of FBG AE Sensor

### 3.1. Packaging Structure

The I-shaped sensor has a relatively simple structure and can effectively protect the grating, but the sensor sensitivity of the bare grating is hardly improved, even causing a decrease of the sensitivity. In addition, the adaptability of range and accuracy to different engineering applications is weak.

According to the analysis and calculation of the packaging structure that is required for engineering applications, the acoustic emission signal is the high-frequency and weak signal, which requires high-sensitivity sensor detection, and the sensitivity-enhancing type packaging is adopted.

Figure 1 shows the two FBG acoustic emission sensor packaging structures that are proposed in this paper, and the structural parameters are optimized. Figure 1a is the general I-shaped fiber grating sensor package structure (PLA-AE-FBG1) and Figure 1b is an improved design (PLA-AE-FBG2) that increases the number of hollowed-out loops. The sensitivity enhancement principle of the sensor structure in the figure is that the fixed distance L2 on both sides is greater than the distance L1 between the adhesives at both ends of the fiber grating. The loop-shape structure in the figure can effectively concentrate the strain in the middle of the package structure. The modulus is greater than that of the fiber so that the deformation $\Delta L_{2}$ of the structure under testing is almost all concentrated on the fiber grating. When the strain of the structure under testing is $\Delta L_{2}/L_{2}$, the strain that is received by the fiber grating is approximately $\Delta L_{2}/L_{1}$. Therefore, the sensitivity amplification factor of the amplifier is $L_{2}/L_{1}$. The purpose of increasing the number of hollows is to further analyze the influence of the sensor structure on the sensitivity.

### 3.2. Simulation Research on Sensitizated Structure

To simplify the experimental objects, we chose simulation research of the loop-shaped package structure close to the physical sensor, and then used the finite element analysis (FEA) method to optimize the modeling of the sensor’s sensitivity enhancement structure in this paper. Static analysis and dynamic analysis are carried out on the general two-loop shaped structure FBG sensor and the improved four-loop shaped structure carvings FBG sensor.

This section mainly studies the strain and response of the grating sensor under continuous stress load and continuous signal excitation.

#### 3.2.1. Static Simulation Analysis

As shown in Figure 2, in the static simulation, one end of the sensor was fixed and restrained and the other end was applied with a continuous load of 1 N-8 N. The direction of the arrow is the direction of load application. The stress distribution cloud diagrams of the four-loop shaped structure and the general two-loop shaped structure that were obtained by the simulation calculation are shown in Figure 3 and Figure 4.

From Figure 3 and Figure 4 and we can see that the strains of the two package structures are mainly concentrated in the hollow area in the middle, which is the area where the grating is located. A continuous load of 1 N–8 N was applied to these two structures, and the maximum strain that was generated on the sensing segment of package structure was recorded. The maximum strain indicates the response sensitivity of the FBG AE sensor. The data is shown in Table 1.

The fitting results of the strain response data curves of PLA-AE-FBG1 and PLA-AE-FBG2 are shown in Figure 5.

In Figure 5, the slopes of the curves are 4.85116 με/N and 3.55957 με/N and the results show that the sensitivity of the four-loop shaped structure of FBG sensor is 1.36 times higher than that of the FBG sensor of two-loop shaped structure. The curves of the two structures have good linearity.

#### 3.2.2. Design of Structural Parameters

We researched the parameters of the sensitization structure by performing simulations. In Figure 6, the dimensions of a, b, and c are used as design parameter. During simulation, one of them is selected as the objective of simulation and increases while the other two values are fixed. This is to make changes to obtain the simulated strain data. We also considered the actual processing conditions and obtained appropriate parameters through data analysis. The relationship between the parameters of a, b, and c and the sensitivity of the sensor was then obtained by analyzing the strain vs. load curve in detail.

This paper uses the four-loop shaped structure model in Figure 1b as the simulation object. First, b and c are fixed as b = 3 mm, c = 4 mm, and parameter is set a to 1 mm, 1.5 mm, and 2 mm. One end of the sensor structure is set as a fixed constraint, and the other end is applied load. With a continuous load of 1–8 N being applied, the change curve of strain is shown in Figure 7; secondly, parameter b is set to 3 mm, 5 mm, and 8 mm, a and c are fixed to a = 2 mm, c = 4 mm, and the above process is repeated to obtain the change trend curve as shown in Figure 7b; finally, a and c are fixed a = 2 mm, c = 3 mm, parameter c is set with values of 4 mm, 5 mm, and 6 mm, and the above process is repeated to obtain the strain trend curve as shown in Figure 7.

The simulation data are shown in Table 2.

The data of the experimental results are fitted so that when the parameter a takes different values, the sensitivity is 4.98 με/N, 6.19 με/N, and 8.93 με/N. From Figure 7a it is shown that the sensitivity increases along with the increase of parameter a, and the more obvious the influence on the sensitivity. When the value of parameter b is different, the corresponding slopes are 8.26 με/N, 6.76 με/N, and 5.52 με/N. From Figure 7b it is shown that the smaller the parameter b is 3 mm, the higher the sensitivity is 4 mm. When the value of parameter c is different, the smaller the parameter at c, the greater the sensitivity, and the sensitivities obtained are 8.93 με/N, 6.61 με/N, and 5.45 με/N, respectively. Therefore, to consider the packaging size of the fiber grating and the relationship between the sensitivity and the packaging parameters, the parameters of the enhanced sensitivity packaging such as a, b, and c have been chosen as 2 mm, 3 mm, and 4 mm, and achieve the optimal sensitivity enhancement effect for the packaging structure of FBG sensor.

#### 3.2.3. Dynamic Excitation Simulation Analysis

The package structures of PLA-AE-FBG1 and PLA-AE-FBG2 were researched in the simulation experiment of dynamic excitation. The simulation of dynamic excitation is shown in Figure 8. The rectangular plate with a size of 20 mm × 70 mm × 2 mm is placed under the package model. The material is set to aluminum. One end of the aluminum plate is set as the excitation input end in the direction of the left arrow in Figure 8. A perfect matching layer is set around the aluminum plate to prevent interface reflections. The wave and the excitation signal are mixed together to produce errors.

The excitation source is a sinusoidal signal; the frequency is set to 150 kHz; the specified displacement amplitude is set to 0.1 mm, 0.5 mm, 1.0 mm, 1.2 mm, and 1.5 mm; and the excitation time is set to 0.1 ms. The displacement of the grating sensing section was studied, and the global point diagram of the grating sensing part with different excitation amplitudes is calculated by simulation where the time of the response data is 0.1 ms. Figure 8 shows the displacement changes of the grating sensing part. The transient state of the grating section was obtained by simulation. The spectrogram was obtained by performing the spectrum analysis on the simulation of the response curve. It can be clearly seen from Figure 9 that the main peak in the frequency spectrum is at 150 kHz, which is the same as the frequency of the excitation source.

The peak-peak values of the dynamic response curve as the response data are shown in Table 3.

The plotting curves of the experimental data under different excitation displacement are shown in Figure 10 for the module structure of PLA-AE-FBG1 and PLA-AE-FBG2.

It can be seen from Figure 10 that the standard signals under different excitations were detected by PLA-AE-FBG1 and PLA-AE-FBG2. The two sensors have good response consistency. The fitting curve equation 5 is obtained from the data in Table 3.
(5)$$\{\begin{array}{c} y_{1}=0.20711x-0.01322\\ y_{2}=0.30404x-0.02545 \end{array}$$

In the above formula, it is demonstrated that the response sensitivity of the two FBG sensors are 0.20711 and 0.30404, respectively. The results show that the FBG acoustic emission sensors have good sensing properties and follow the excitation signal curve. The dynamic response sensitivity coefficient of PLA-AE-FBG2 is 1.47 times higher than that of PLA-AE-FBG1. Through the simulation analysis of the FBG sensor’s sensitivity-enhancing structure, the four-loop shaped structure of the FBG AE sensor has good dynamic response characteristics.

### 3.3. Sensor Manufacture

An additive manufacturing method and FBG sensing technology were combined to fabricate a small size sensor which has advantages of high precision, high repeatability, rapid production, and low cost. It has been gradually used in the field of optical fiber sensing. Currently, the printing materials of commercially non-metallic include polylactic acid (PLA), nylon, and acrylonitrile butadiene styrene (ABS), etc. By comparing references [1,2], PLA was finally selected as the printing material. Among them, PLA has good thermal stability, excellent strength and ductility, and good corrosion resistance. Therefore, PLA was used as the packaging material of FBG in this paper. According to the ASTM D6272 standard for bending experiments, and it was shown that PLA has higher rigidity than ABS [35]. The parameters of the 3D printing process should be adjusted, such as setting the extrusion temperature and the printing speed to 220 °C and 30 nm/s, respectively [32].

Through the simulation result, the sensor package structure was determined: the number of loop-shapes is 4, the values of a, b, and c are 2 mm, 3 mm, and 4 mm, respectively. Under the load of 8 N, the sensor thickness is 1.5 mm, 2 mm, 2.5 mm, and 3 mm, respectively. The response strain variation curve is shown in Figure 11 through simulation analyzing.

According to the analysis of the results, the sensitivities of the different structural model thicknesses are 71.3966 με/N, 50.016 με/N, 36.6767 με/N, and 31.6784 με/N, demonstrating that the larger the sensor package thickness, the lower the strain sensitivity. According to the experimental results, the final thickness parameter of the FBG acoustic emission sensor is set to 1.5 mm, and the thickness of the adhesive base of the package structure is 0.5 mm.

In the experiment, the FBG grating is produced by Shandong Shenghai Optical Technology. The length of grating area is 10 mm, the reflectivity is 93.27%, the 3 dB bandwidth is 0.24, the central wavelength is 1553.054 nm, and the fixing point of the fiber grating is encapsulated and fixed with epoxy resin glue. The center wavelength of the fiber grating is 1552.880 nm before encapsulation. After curing the epoxy resin for 24 h, the center wavelength after encapsulation becomes 1552.960 nm due to the pre-stress that is introduced during the encapsulation process. The photograph of the AM-packaged FBG AE sensor is shown in Figure 12.

## 4. Experiments and Discussion

According to the FBG sensing principle, we built the FBG acoustic emission wave sensing system based on the FBG AE detection test combining the actual requirements of AE wave detection on the aluminum plate. In this setup, the experimental setup for AE wave using FBG AE sensor is shown in Figure 13, which consists of the fiber grating AE demodulation system and an FBG AE sensor. The tunable narrowband light penetrates the optical circulator and enters the FBG AE sensor. The reflected light of FBG AE sensor that contains the AE wave signal enters the high-speed photodetector and is converted into the electrical signal, which finally reached the oscilloscope and is displayed by the oscilloscope.

To verify the performance of the four-loop shape carvings of the FBG AE sensor, the acoustic wave was generated by pencil lead break and continuous PZT sine pulse respectively in this experiment.

During measuring, the narrowband tunable light source uses Santec TLS-550, the output power is set to 5 mW, and the output wavelength was set to 1552.800 nm (we chose the 20–80% linear segment of the left front of the center wavelength), so that it is at the left edge of the sensor grating spectrum 3 dB bandwidth. The photodetector uses Newport 2053, and the digital oscilloscope uses Keysight DXO-X3024. The aluminum plate thickness was 15 mm for the test. The arbitrary function generator was used to drive the AE probe for generating the continuous AE simulation signal, thereby generating stress waves in the aluminum plate. By adjusting the light source output to 3 dB of the fiber grating, it could ensure that the grating works in the linear region and achieves the maximum sensitivity. We attempted to keep the laboratory temperature constant at 20 °C during the test.

### 4.1. Experiment Study on Pencil Lead Break Signal Response

In the experiment, bonding the PZT AE sensor, the PLA-AE-FBG sensor is fixed on the aluminum plate with α-ethyl cyanoacrylate. The pencil lead break signal is used as the standard AE signal source. The PZT AE sensor (RS54a) and PLA-FBG-AE sensor are collected on the oscilloscope at the same time. The collected data are shown in Figure 14.

Figure 14 presents the response signal of the PLA-AE-FBG2 sensor, AE-PZT sensor and PLA-AE-FBG1 sensor from top to bottom. The experimental results show that the peak-peak values of PLA-AE-FBG1 and PLA-AE-FBG2 are Vpp = 548 mv and Vpp = 806 mv, respectively, indicating that the sensitivity coefficient of PLA-AE-FBG2 is 1.47 times higher than that of PLA-AE-FBG1, and that of AE-PZT has a peak-peak value Vpp = 1112.9 mv, which is higher than the FBG AE sensor. Good consistency was achieved among the three sensors for the pencil lead break signal. It shows that both the PLA-AE-FBG sensor and the AE-PZT sensor have good AE sensing characteristics about the application of AE detection.

### 4.2. Experiment Study on Continuous Signal Response

Figure 15 shows how the sensors are laid on the aluminum plate. The PZT AE probe (R15a) is excited by a signal generator, which is as the standard continuous AE signal to measure the stability of the FBG AE sensor.

We attached the PLA-AE-FBG2 and PLA_AE-FBG1 sensor on the aluminum plate with dimensions of 1000 mm × 1000 mm × 15 mm and placed the RS54a (AE-PZT) next to the R15a. The distance between the sensors and AE source was 5 cm. The output frequency of the function generator was set to 150 kHz, the Vpp is set to 20 V, 15 V, 10 V, and 5 V, respectively. The continuous AE signal was emitted by the R15a, which propagated in the aluminum plate, and then reached the PLA-AE-FBG sensor and the AE-PZT sensor. The output power of the narrowband laser light source was set to 5 mW, the center wavelength was set to 1552.8 nm which was at the full width half maximum (FWHM) of the reflection spectrum. The adjusted laser signal passes through the circulator to enter the PLA-AE-FBG sensor, the reflected light of the FBG sensor is converted into an electrical signal by a photodetector (Newport 2053). The bandpass filter that was embedded in the photodetector was set to 100 kHz–300 kHz, the electrical signal can effectively filter out the interference of other frequency bands after passing through the bandpass filter signal. The output electrical signals of the PZT AE sensor and FBG AE sensor were directly connected to the oscilloscope for real-time measurement, display, and storage. The small spectral shifts that were caused by high frequency AE waves could be converted to light intensity variations. Therefore, the high sensitivity of AE sensor is realized for this sensing system. The spectrum curve is acquired from performing spectrum analysis on the data that were collected from the oscilloscope, as shown in Figure 16.

The experimental main response frequency of the spectrum is 150.02 kHz, which agrees very well with the excitation sinusoidal signal. The results demonstrate that the designed sensitization structure of the FBG AE sensor can respond well to the AE signal.

During the experiment, the frequency of the function generator signal is kept unchanged at 150 kHz, and the amplitude of the signal excitation is set to 20 V, 15 V, 10 V, and 5 V to generate continuous AE signals. Under different signal amplitude excitation conditions, the response curves of the PLA-AE-FBG sensors and the AE-PZT are as follows as shown in Figure 17.

The analysis of the experimental results shows that under different excitations, the PLA-AE-FBG1 and PLA-AE-FBG2 sensors can respond correctly to the excitation signal as well as the PZT AE sensor and has good stability. The response data fitting of the PLA-AE-FBG1 and PLA-AE-FBG2 are performed on the peak-peak value, when the peak-peak value of the excitation signal is raised from Vpp = 5 V to Vpp = 20 V. The experiment process was repeated two times. The corresponding data were used to conduct fitting curves as shown in the Figure 18.

As shown in Figure 18, when the excitation signal value is raised from 5 V to 20 V, the fitting curve is basically the same. From the fitting results, the sensitivity coefficient of PLA-AE-FBG2 sensor is 3.057, and that of PLA-AE-FBG1 is 2.0702. The sensitivity coefficient of PLA-AE-FBG2 is 1.475 times that of PLA-AE-FBG1, which is basically consistent with the simulation. When the Vpp is different, the signal that is detected by the PLA-AE-FBG sensor has good linearity and repeatability, which can effectively ensure the stability of the FBG sensor.

## 5. Conclusions and Future Study

In this paper, a four-loop shaped carving structure of an FBG AE sensor packaging is proposed. The effect of the structure parameters of the FBG AE sensors on the sensitivity is studied by using finite element simulation analysis. The main parameters of the enhanced sensitivity structure are a = 2 mm, b = 3 mm, and c = 4 mm, and the number of loop-shape carvings is four. In the FBG AE sensor response experiment, the results demonstrate that the designed sensitization structure of the FBG AE sensor can respond well to the pencil lead break and the continuous sine signal. It is verified that the PLA-AE-FBG2 sensor is compatible with the PZT AE sensor, and has good acoustic emission sensing characteristics, which can realize the detection of AE signals. The study shows that the designed the four-loop shaped structure of the FBG AE sensor effectively improves the sensitivity of the FBG sensor, and the sensitivity factor is increased by 1.47 times. The four-loop shaped structure has the sensitive sensing characteristics, good linearity, and stability of the AE signal, as well as a higher rigidity which enhances the application reliability and application range. It is possible to apply the FBG AE sensor in some complex engineering environments.

Future studies need to be conducted on the miniaturization of the FBG AE sensor and the distributed sensor network to improve the rate of identification and positional accuracy of acoustic emission sources. To solve the application problem of on-line acoustic emission detection of composite materials and aero-engine blade defects, a high temperature resistance of FBG-AE sensor should be designed to identify blade defects and predict the life of blade.

## Figures and Tables

**Figure 1 sensors-22-00416-f001:**
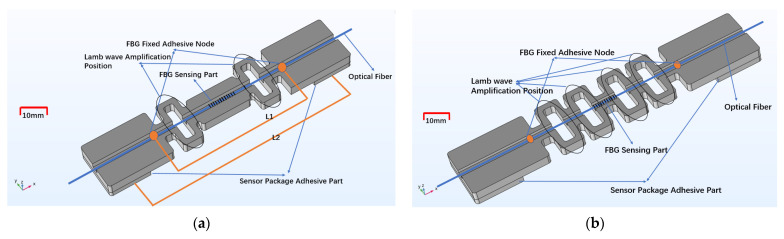
The loop-shape package structures. (**a**) Normal I-shaped FBG sensor model; (**b**) Improved I-shaped FBG sensor.

**Figure 2 sensors-22-00416-f002:**
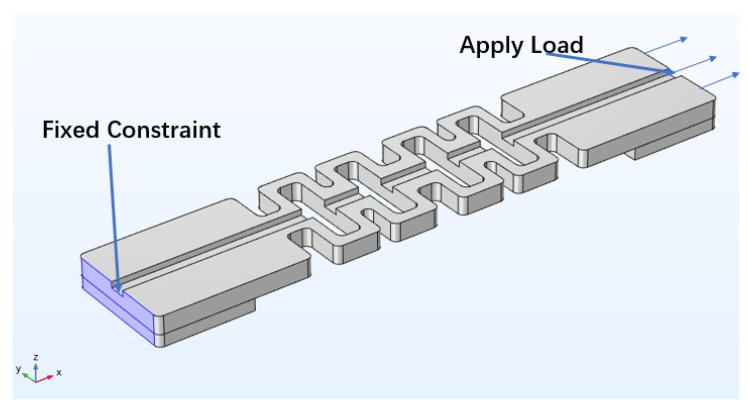
Structural simulation model. The highlighted region represents the fixed constraint. The arrow is the direction of load application.

**Figure 3 sensors-22-00416-f003:**
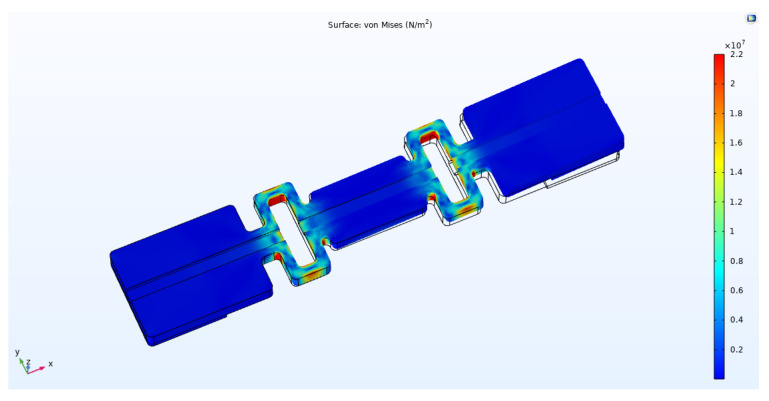
Stress contour plots of PLA-AE-FBG1.

**Figure 4 sensors-22-00416-f004:**
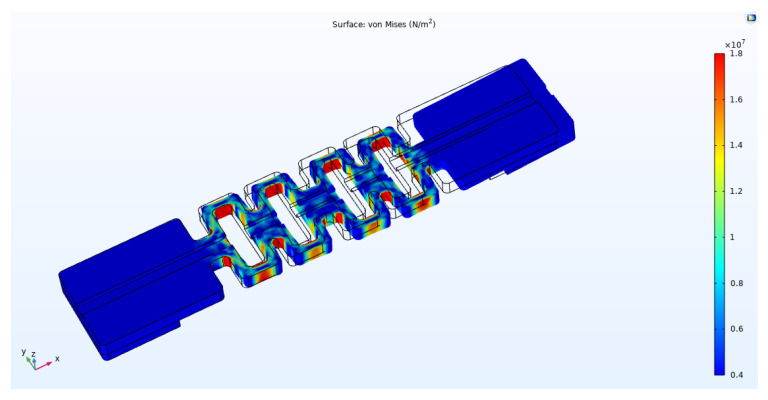
Stress contour plots of PLA-AE-FBG2.

**Figure 5 sensors-22-00416-f005:**
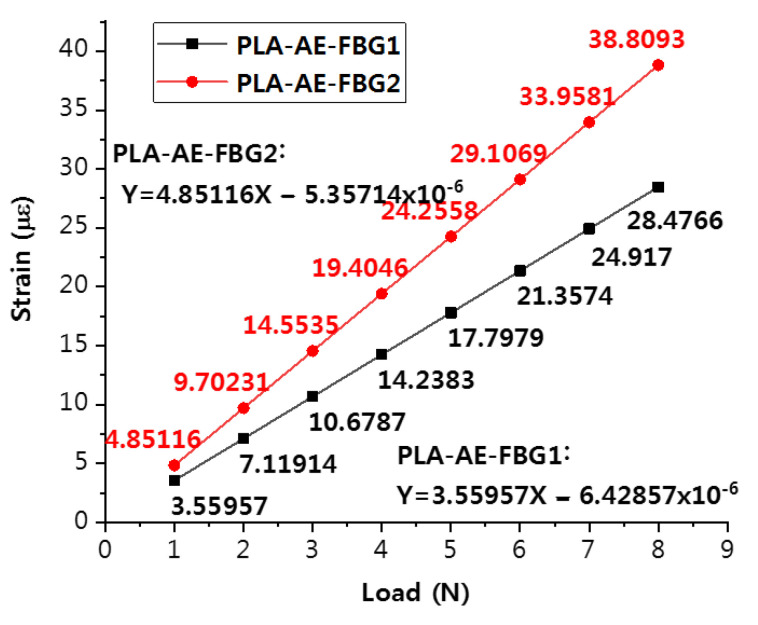
Stress response curves of different structures under the continuous load.

**Figure 6 sensors-22-00416-f006:**
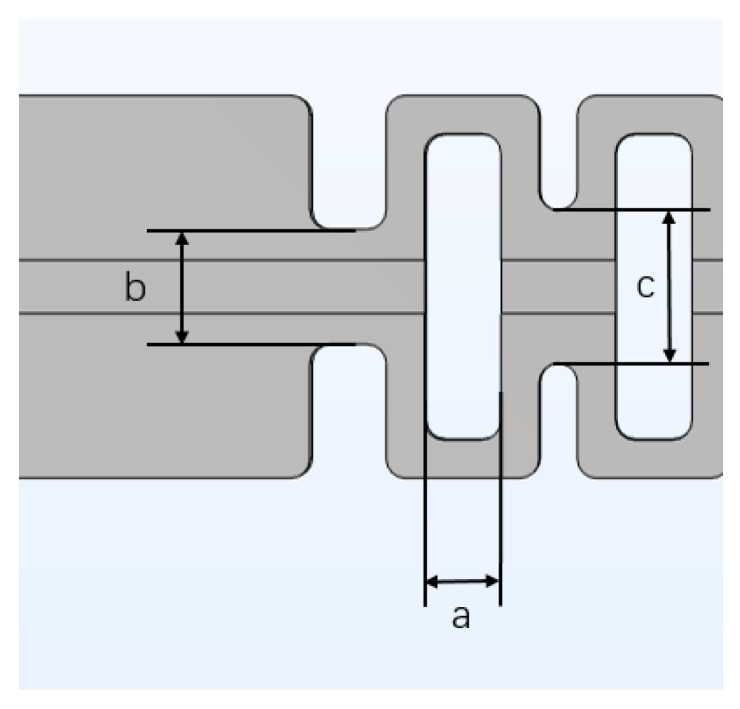
Parameters of the strain amplification area of the sensor package.

**Figure 7 sensors-22-00416-f007:**
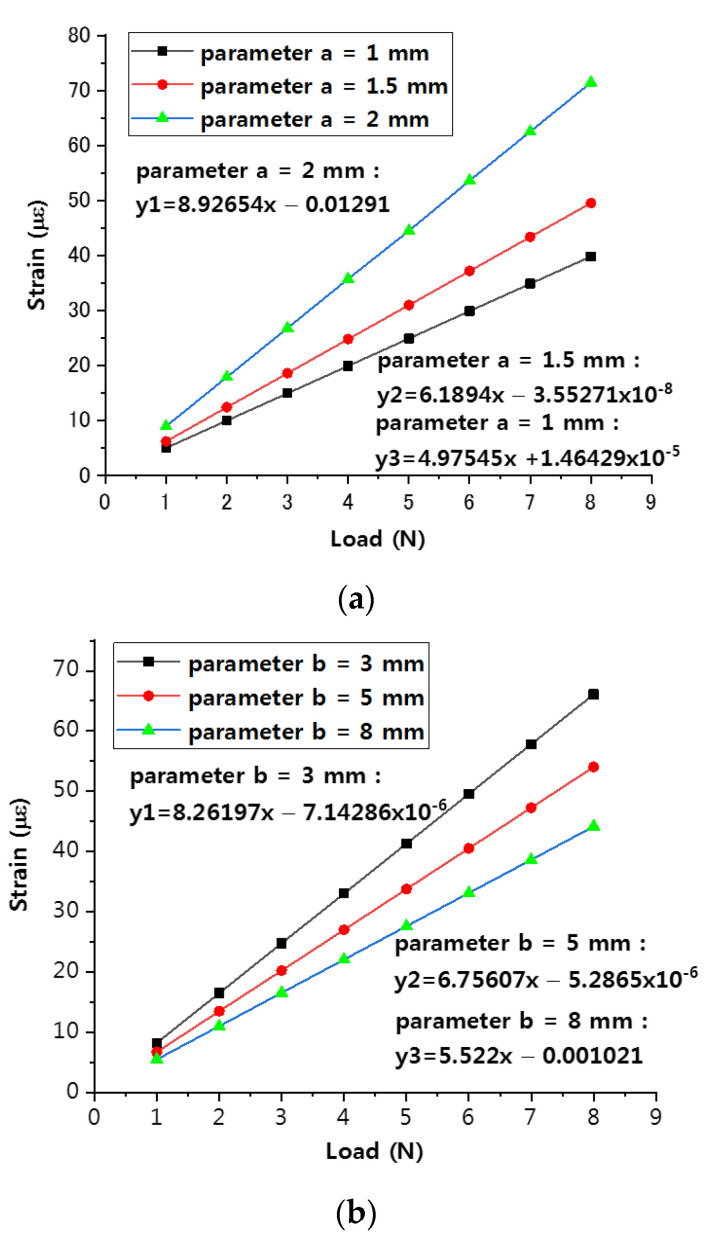
Influence curve of the different parameters in the amplification area. (**a**) Influence of parameter ‘a’ to strain; (**b**) influence of parameter ‘b’ to strain; (**c**) influence of parameter ‘c’ to strain.

**Figure 8 sensors-22-00416-f008:**
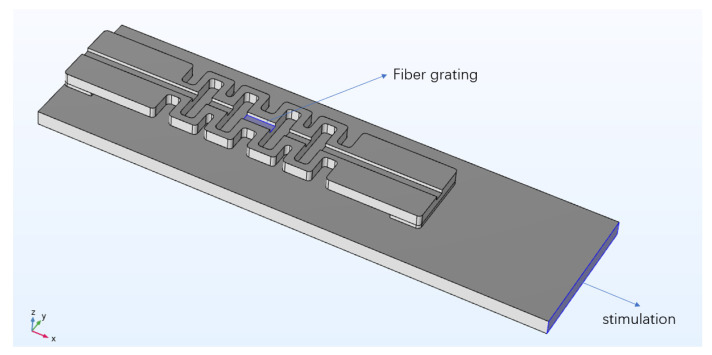
Dynamic excitation loading diagram. The highlighted region represents the stimulation and fiber grating.

**Figure 9 sensors-22-00416-f009:**
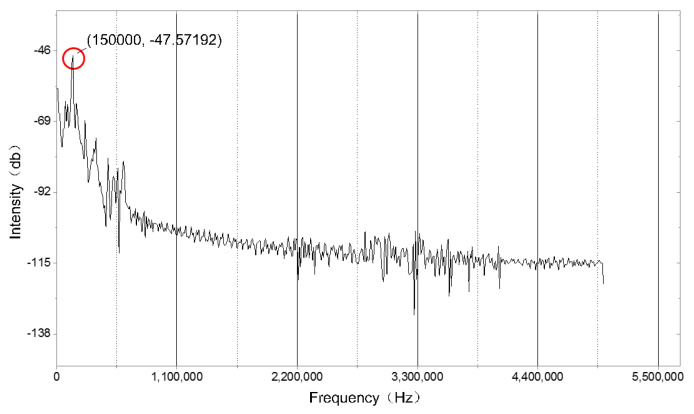
Spectrum of the simulated dynamic response signal.

**Figure 10 sensors-22-00416-f010:**
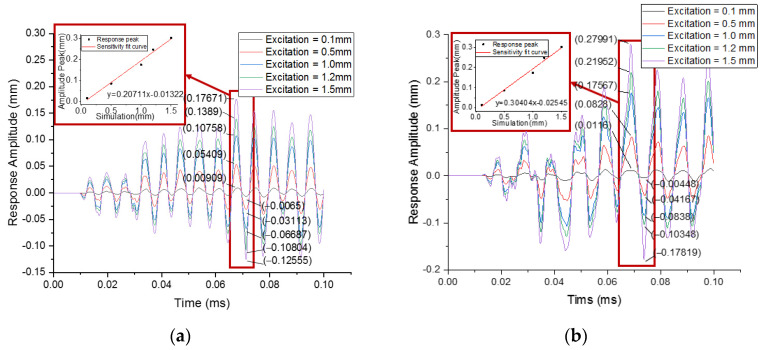
The response curve under different excitation signals. (**a**) PLA-AE-FBG1 structural response curve; (**b**) PLA-AE-FBG2 structural response curve.

**Figure 11 sensors-22-00416-f011:**
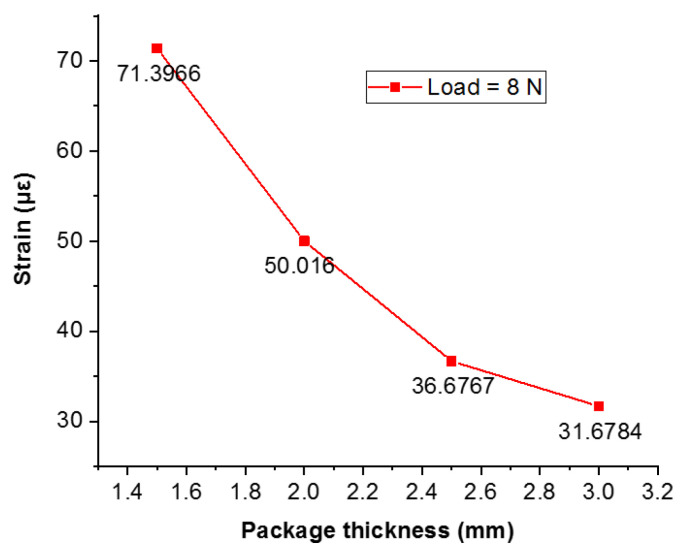
Strain curve diagram of the sensor packaging thickness.

**Figure 12 sensors-22-00416-f012:**
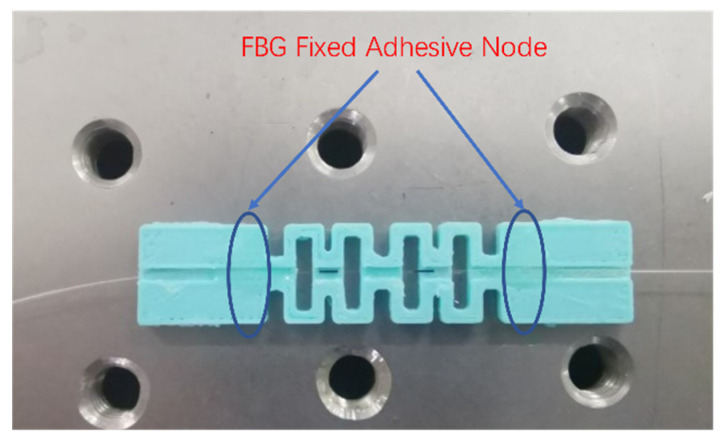
Picture of PLA-AE-FBG2 sensor.

**Figure 13 sensors-22-00416-f013:**
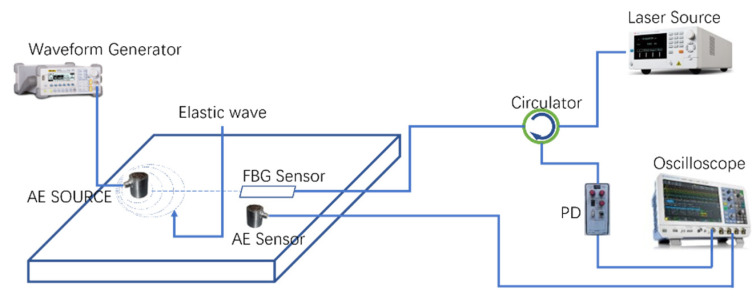
Schematic diagram of AE detection system.

**Figure 14 sensors-22-00416-f014:**
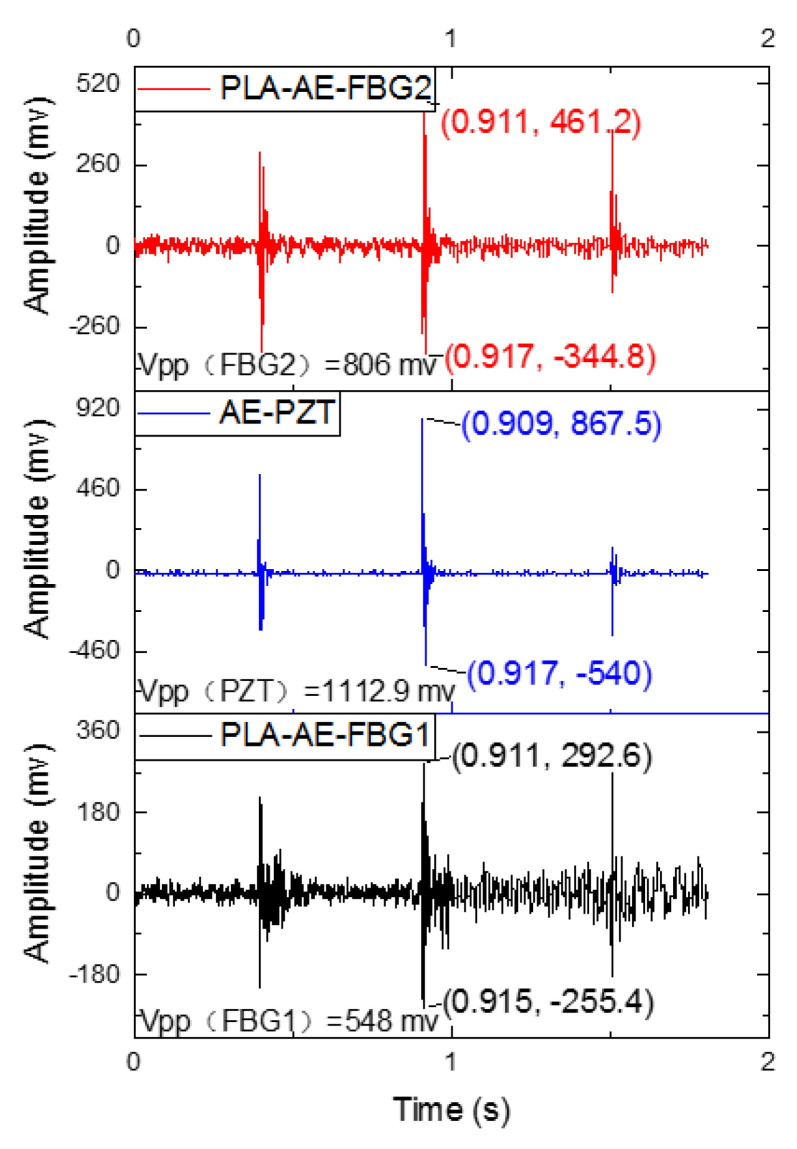
Amplitude-time curves.

**Figure 15 sensors-22-00416-f015:**
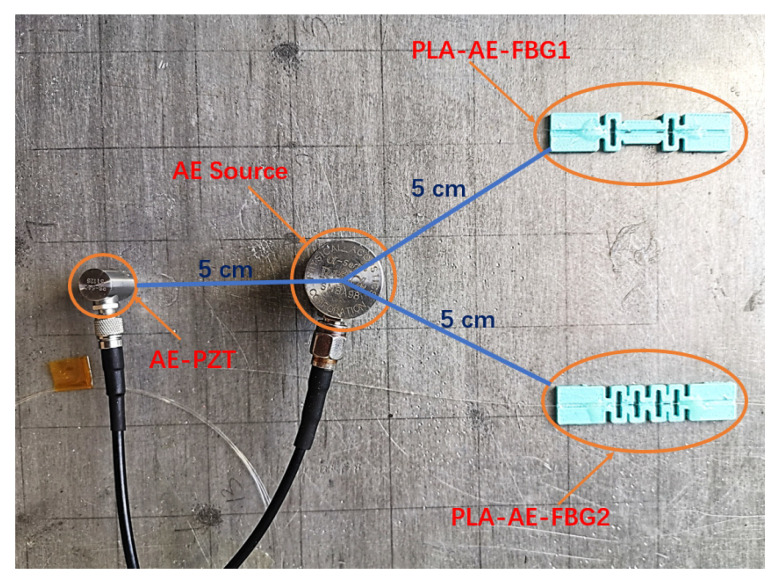
Experimental diagram of continuous AE signal detection.

**Figure 16 sensors-22-00416-f016:**
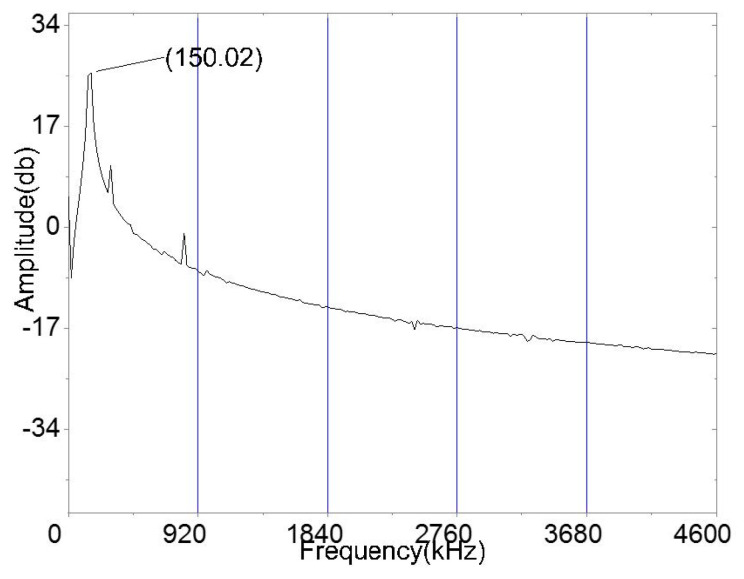
Amplitude-frequency curves.

**Figure 17 sensors-22-00416-f017:**
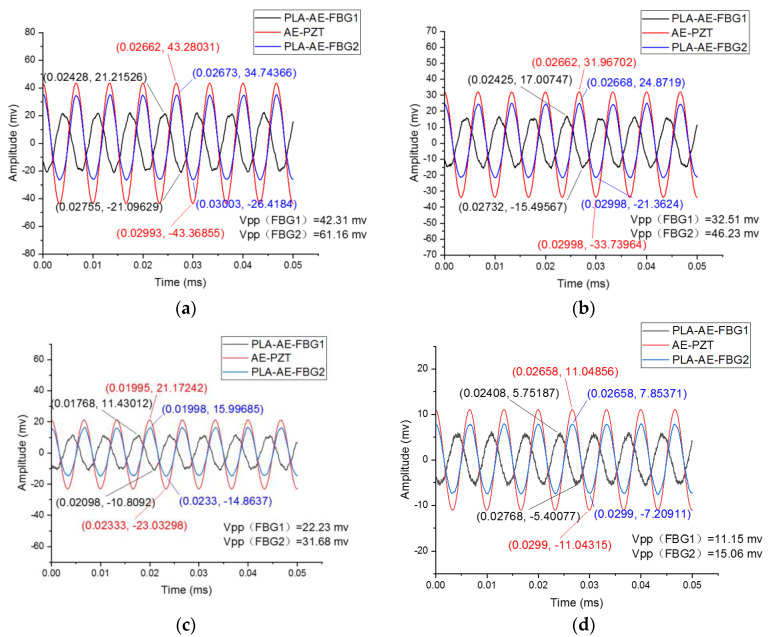
The comparison of the FBG AE sensor and the PZT AE sensor under different amplitude excitations (**a**) The peak-peak value of the excitation signal is Vpp = 20 V; (**b**) The peak-peak value of the excitation signal is Vpp = 15 V; (**c**) The peak-tpeak value of the excitation signal is Vpp = 10 V; (**d**) The peak-peak value of the excitation signal Vpp = 5 V.

**Figure 18 sensors-22-00416-f018:**
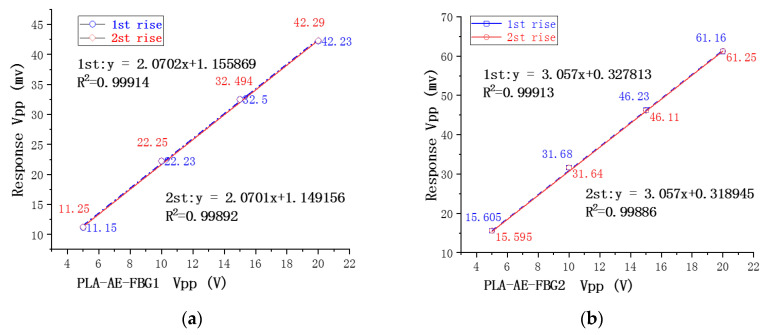
Fitting the data curve under different amplitude excitations (**a**) When the peak-peak value of the excitation signal is raised from 5 V to 20 V, the fitting data curve of PLA-AE-FBG1; (**b**) When the peak-peak value of the excitation signal is raised from 5 V to 20 V, the fitting data curve of PLA-AE-FBG2.

**Table 1 sensors-22-00416-t001:** Strain response data of PLA-AE-FBG1 and PLA-AE-FBG 2 under continuous load.

LOAD	PLA-AE-FBG1 (με)	PLA-AE-FBG2 (με)
1 N	3.55957	4.85116
2 N	7.11914	9.70231
3 N	10.6787	14.5535
4 N	14.2383	19.4046
5 N	17.7979	24.2558
6 N	21.3574	29.1069
7 N	24.917	33.9581
8 N	28.4766	40.8093
Fitting curve slope	4.244768	3.114692

**Table 2 sensors-22-00416-t002:** The data of parameters a, b and c under different loads.

Parameter (mm)		Load = 1 N	Load = 2 N	Load = 3 N	Load = 4 N	Load = 5 N	Load = 6 N	Load = 7 N	Load = 8 N	Fitting Curve Slope
a	1	4.97545	9.95091	14.9264	19.9018	24.8773	29.8527	34.8282	39.8036	4.97545
	1.5	6.1894	12.3788	18.5682	24.7576	30.947	37.1364	43.3258	49.5152	6.1894
	2	8.9286	17.8573	26.786	35.7147	44.4634	53.572	62.5007	71.4294	8.92654
b	3	8.262	16.5239	24.7859	33.0479	41.3099	49.5718	57.8338	66.0958	8.26197
	5	6.7561	13.5121	20.2682	27.0243	33.7804	40.5364	47.2925	54.0486	6.75607
	8	5.522	11.044	16.566	22.088	27.61	33.132	38.654	44.176	5.522
c	4	8.9286	17.8573	26.786	35.7147	44.4634	53.572	62.5007	71.4294	8.92654
	5	6.6123	13.2246	19.8368	26.4491	33.0614	39.6737	46.2839	52.8982	6.61215
	6	5.4434	10.8867	16.3301	21.7735	27.2168	32.8602	38.1036	43.5469	5.45051

**Table 3 sensors-22-00416-t003:** The response data of PLA-AE-FBG sensor under different excitation signals.

Sensor	Vpp = 0.1 mm	Vpp = 0.5 mm	Vpp = 1.0 mm	Vpp = 1.2 mm	Vpp = 1.5 mm
PLA-AE-FBG1	0.01559	0.08522	0.17445	0.24694	0.30226
PLA-AE-FBG2	0.01608	0.12547	0.25947	0.323	0.4561

## Data Availability

The study did not report any data.

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
