# Peer review of "Design Optimization of Sensitivity-Enhanced Structure for Fiber Bragg Grating Acoustic Emission Sensor Based on Additive Manufacturing"

_sensors, 2022, doi:10.3390/s22020416_

Round 1
Reviewer 1 Report
The manuscript “
Design Optimization of Sensitivity-Enhanced Structure for Fi-2 ber Bragg Grating Acoustic Emission Sensor Based on 3D Printing” dealing with 3dprinting of MEX has been refereed. Please comments are listed below. The paper needs major revision. I encourage the author to read and address my comments:
- Use ASTM 52900 for correct terminologies of AM processes.
- Add some quantitative results to the abstract.
- Additive manufacturing now has many advantages over conventional manufacturing. To highlight your work, add a short note in the introduction by using the following paper and mention the privilege of additive manufacturing. “Additive manufacturing a powerful tool for the aerospace industry” 2021.
- The contribution of the paper is not clear in the current format. Highlight the contribution of the paper and make it bolder.
- Figure 1 has no point for this paper. Please remove it.
- What are the future directions of this research? Authors are encouraged to write the future direction.
- Update the introduction by comparing the following new references on PLA printing with reviewed paper in your article.
- Kumar Mishra, P., S. Ponnusamy, and M.S. Reddy Nallamilli, The influence of process parameters on the impact resistance of 3D printed PLA specimens under water-absorption and heat-treated conditions.
- von Windheim, N., D.W. Collinson, T. Lau, L.C. Brinson, and K. Gall, The influence of porosity, crystallinity and interlayer adhesion on the tensile strength of 3D printed polylactic acid (PLA)
- Afonso, J.A., J.L. Alves, G. Caldas, B.P. Gouveia, L. Santana, and J. Belinha, Influence of 3D printing process parameters on the mechanical properties and mass of PLA parts and predictive models
- Gonzalez Alvarez, A., P.L. Evans, L. Dovgalski, and I. Goldsmith, Design, additive manufacture and clinical application of a patient-specific titanium implant to anatomically reconstruct a large chest wall defect.
- Travieso-Rodriguez, J.A., R. Jerez-Mesa, J. Llumà, G. Gomez-Gras, and O. Casadesus, Comparative study of the flexural properties of ABS, PLA and a PLA–wood composite manufactured through fused filament fabrication.
Reviewer 2 Report
The authors present an interesting research about acoustic emission sensor based on 3D printing and embedded FBGs to improve the sensitivity. A detailed document is well presented and discussed. I ahve some comments to be addressed.
- Introduction: please improve the part when is mentioned "Fiber grating sensor has the advantages of rapid response, anti-electromagnetic interference, small size, easy installation, high reliability, etc. The sensor network of distributed or quasi-distributed is easier to setup." I miss some references to highlight this part as potential simultaneous measurement when it is used single FBG element, for instance: Journal of Lightwave Technology 37 (3), 971-980, 2019; Results in Optics, 5, 2021, 100135.
- There is no details about the FBG fabrication and details about physical length of FBG, intensity, etc. Also, from Fig. 2 seems the FBG design, in terms of physical length for normal I-shaped, more long than improved I-shaped. What is the influence to use different physical FBG length? Or do you used the same FBG length for a correct comparison between both configurations (a and b)? Please be clear about it on the document and also about details of FBG 1 and FBG2.
- Improve some legends of figures. Some are quite basic.
- Fig. 16: is not clear about the distance of AE source and sensing elements. Also the read color in the figure is not the best color to read.
- Fig. 19: how many times was repeated the cycle? just one? please add more data for repeatability test. Also, Y1 and Y2 are not identified in a correct way in figure.
- ANy type of fixation between the sensing elements to the optical table? What is the influence on it? Please comment.
Round 2
Reviewer 1 Report
The paper is ready to go.
Reviewer 2 Report
The paper was improved as suggested and clarified and can be published.